# State of Charge Estimation of Lithium-Ion Batteries Using Stacked Encoder–Decoder Bi-Directional LSTM for EV and HEV Applications

**DOI:** 10.3390/mi13091397

**Published:** 2022-08-26

**Authors:** Pranaya K. Terala, Ayodeji S. Ogundana, Simon Y. Foo, Migara Y. Amarasinghe, Huanyu Zang

**Affiliations:** Department of Electrical and Computer Engineering, Florida A&M University-Florida State University, 2525 Pottsdamer St., Tallahassee, FL 32310, USA

**Keywords:** machine learning, energy storage, state-of-charge estimation, bi-directional LSTM, encoder–decoder hybrid, robust estimator, deep neural network

## Abstract

Energy storage technologies are being used excessively in industrial applications and in automobiles. Battery state of charge (SOC) is an important metric to be monitored in these applications to ensure proper and safe functionality. Since SOC cannot be measured directly, this paper puts forth a novel machine learning architecture to improve on the existing methods of SOC estimation. This method consists of using combined stacked bi-directional LSTM and encoder–decoder bi-directional long short-term memory architecture. This architecture henceforth represented as SED is implemented to overcome the nonparallel functionality observed in traditional RNN algorithms. Estimations were made utilizing different open-source datasets such as urban dynamometer driving schedule (UDDS), highway fuel efficiency test (HWFET), LA92 and US06. The least Mean Absolute Error observed was 0.62% at 25 °C for the HWFET condition, which confirms the good functionality of the proposed architecture.

## 1. Introduction

In the 21st century, the global automobile industry is moving on to electric vehicles. Government-backed policies such as the United States government pledging a 50% reduction in transport sector CO_2_ emissions by 2030 has given a more legitimate reason for a switch to electric vehicles and has led to an ever-growing demand for hybrid and electric vehicles [1,2]. The automobile racing industry has also made the switch from highly inefficient V8 power systems to more efficient hybrid V6 engines in formula 1 championship and fully electric drive systems in formula E championship. With the increased demand for electric and hybrid electric vehicles, manufacturers are looking into ways to improve and accurately monitor and estimate various parameters such as the state of charge and state of health of the battery packs being used in these vehicles. A Hybrid Electric Vehicle (HEV) has an electric motor working in conjunction to a conventional engine; because of this, HEVs achieve better fuel economy. Fuel economy in plug-in electric vehicles (PHEV) and all electric vehicles (EV) is calculated differently because of the apparent change in the propulsion systems. The two common metrics for fuel economy in HPEVs and EVs is miles per gallon of gasoline equivalent (mpge) and kilowatt-hours (kWh) per 100 miles. According to the United States Department of Energy, the 2018 Accord hybrid has an EPA combined city and highway fuel economy of 47 miles per gallon, while on the other hand, the estimate for a Honda Accord (petrol) with a four-cylinder internal combustion engine is 33 miles per gallon. 

Unlike a conventional car where fuel gauging is completed using a float connected to a resistor in the fuel tank and fuel is gauged depending on the position of the float, it is apparent that this system of fuel gauging is not useful in an electric car. Hence, electric vehicles employ state of charge (SOC) estimation to determine the charge remaining in the battery pack. SOC estimation is therefore also known as the “gas gauge” or “fuel gauge’ function.

### 1.1. Related Work

Data-driven SOC estimation techniques have gained a lot of traction in recent years because they do not require a complex battery model and only require battery data for SOC estimation. Neural networks have been used previously for SOC estimation [3], and hybrid neural network architectures such as neural networks combined with extended Kalman filtering for error cancellation have also been proposed [4]. Other adaptive methods using back propagation have also been proposed with Shen et al. proposing a SOC estimation technique using a neuro-controller based on back propagation [5]. Dong et al. used an improved back propagation neural network [6] which was an improvement over the back propagation NN proposed by Sun et al. [7]. These studies lay a good foundation for neural network-based SOC estimation but lack the high accuracy required for HEV applications. Due to the advances in computers and increased accessibility to more powerful machine learning workbenches, deep neural networks can be employed for SOC estimation. 

Deep neural networks (DNN) are a branch of machine learning that uses multiple hidden layers as compared to a few hidden layers used in neural networks. Deep neural networks help in achieving the high accuracy levels demanded by modern SOC estimation techniques and are ideal when dealing with a high volume of data [8]. Studies based on layer stacking and SOC estimation using multiple layer perceptron have been proposed [9,10]; these studies show the advantages of using DNNs in SOC estimation. Chemali et al. have further improved on the multilayer perceptron model by using long short-term memory (LSTM) units and achieved good accuracy [11]. LSTM units have also been used by Addas et al. for SOC estimation and achieved similar accuracy [12]. Gated recurrent units can be used to substitute the LSTM units, Zhao et al. used GRUs to model the non-linear behavior of batteries and proposed different input models for SOC estimation [13]. Other studies have also used GRUs to estimate SOC at fixed ambient temperatures [14,15]. Chen et al. was able to obtain good accuracies while using GRU-RNN network with a GRU-ATL activation function layer [16]. Although accurate SOC estimates can be made, most of these studies estimate SOC at fixed ambient temperatures and do not exploit the backwards dependencies in the battery data. Bi-directional LSTM can be employed to take advantage of the bi-directional temporal dependencies in a time series data [17]. Stacked Bi-LSTM and encoder–decoder Bi-LSTM have been previously proposed for SOC estimation at varying ambient temperatures [18,19]. Although these networks provide a reliable and stable SOC estimation, more accurate SOC estimation is required for HEV applications and can be achieved by using hybrid architectures. 

To address the mentioned limitations, a new stacked encode–decoder network is proposed in this study that aims to improve the estimation accuracies by combining stacked and encoder–decoder architectures. Most of the studies mentioned use a standard training and testing database which contain different drive cycle data at various ambient temperatures. In this study, the same database is used with varying ambient temperatures. This makes the SOC estimation a sequence-to-sequence regression task because the input measurement sequences must be mapped to the corresponding output SOC values. The proposed network works well for this type of sequence-to-sequence regression task. The use of Bi-LSTM units provides the advantage of the network’s ability to learn bi-directional temporal dependencies, which helps in a more accurate and stable SOC estimation. The use of encoder–decoder architecture, where both the encoder and decoder blocks are trained simultaneously, allows the network to capture the sequential patterns more efficiently within an input time series [18]. Given the relative ease with which a DNN network can be implemented for BMS applications, the proposed network offers a novel solution [20]. The contributions of this paper are:(1)A novel hybrid architecture using a stacked Bi-LSTM and encoder–decoder Bi-LSTM is used to estimate SOC at varying temperatures. The network can take advantage of the bi-directional functionality of Bi-LSTMs and capture sequential tendencies more accurately and provide a more accurate SOC sequence. By providing a SOC sequence estimate as opposed to single value SOC estimates, the trend in battery capacity and battery state in real-world scenarios can be more effectively monitored.(2)The stacked Bi-LSTM was built with deep structures to take advantage of deep neural network architectures, and the use of Bi-LSTM units aid in capturing the temporal dependencies from the forward and backwards directions. Since the encoder and decoder blocks are trained simultaneously, the training time of the network is also reduced.(3)The model is tested on a standard open-source lithium-ion battery dataset. The proposed network performs better than similar pre-existing architectures. Experimental testing proves that the SED network can accurately estimate SOC sequence at varying temperatures provided current, voltage and temperature measurement sequences. A mean absolute error (MAE) of 0.62% was observed for HWFET conditions at varying ambient temperatures, which shows the proper functionality of the proposed network.

The paper is structured as follows, Section 2 describes the LIB battery data and various performance metrics used in the study. Section 3 introduces the proposed network architecture and basic principles of stacked Bi-LSTM and encoder–decoder architectures. Section 4 provides a detailed analysis of the performance of the proposed network, and concluding remarks are given in Section 5.

### 1.2. Battery Management Systems

A BMS is defined as a system that monitors and controls various parameters indicative of battery state of health such as state of charge, instantaneous available power, battery capacity, etc. A BMS is essential to obtain the maximum efficiency from the battery pack in terms of maximum charge available and battery life. A BMS must perform a set list of tasks every measurement interval for proper functionality of the battery pack; these tasks include SOC estimate, SOH estimate, maximum charge available calculation and cell equalization. Figure 1 illustrates a basic BMS task flow with various methods of SOC estimation techniques shown that could be used. 

Most BMS modules use real time voltage (V), current (I) and temperature (T) measurements to perform the task mentioned above. General topologies used to implement BMS architecture are centralized, distributed and modular [21]. 

### 1.3. State of Charge Estimation Techniques

The need for accurate state of charge (SOC) estimation is paramount in HEV and EV applications. Battery state of charge can be defined as the amount of charge left in a battery; unlike a gas-powered car, the remaining charge in the battery cannot be measured directly but has to be estimated using algorithms that predict the available power based on data coming from voltage, current and temperature sensors. Battery SOC can be estimated using different methods, the basic and simple algorithm that can be used is the Coulomb counting method [22,23], where the total amount of energy entering and leaving the battery is monitored. This is implemented using the following equation
(1)$$Z\left(t\right)=Z\left(0\right)-\int_{0}^{t}\frac{\eta_{i}\cdot i\left(t\right)}{C_{n}}dt$$
where Z(0) is initial SOC, $\eta_{i}$ is Coulombic efficiency, i(t) is current and $C_{n}$ is battery capacity. The Coulomb counting method provides an accurate SOC estimate, but to achieve this, a very precise initial SOC estimate is required. Since we are integrating current over time, any current sensor error is accumulated over time, leading to a massive drop in accuracy. 

Other SOC estimation techniques such as fuzzy logic-based SOC estimation [24,25,26], open circuit voltage method [27] and impedance spectroscopy-based methods [28] can be used, but they suffer from issues such as requiring a long time required to obtain OCV measurements; different battery chemistries, temperatures and age affect the OCV [20]. Impedance spectroscopy models differ for different battery chemistries and are highly dependent on experimental conditions [29]. Support vector machine can also be used for SOC estimation [30,31,32]. The lack of robustness in these methods makes them not so good candidates for good SOC estimation algorithm.

Another estimation technique that can be employed other than Artificial Neural Network (ANN) models is battery model based SOC estimation techniques such as Kalman filtering [33], extended Kalman filtering (EKF) [34], etc. The EKF method provides a very precise and robust SOC estimate [35]. Although methods such as EKF provide accurate SOC estimates, they are highly dependent on a very accurate battery model; for instance, in [36], a Kalman filtering example is provided using a simple linear circuit that is considered as a model of the battery, but for extremely precise estimates of SOC, a far more complicated and accurate battery model is required [37]. Apart from the dependency on accurate battery models, the EKF method requires a high computational cost after integration compared to other methods [20].

To overcome these challenges and provide an equally precise SOC estimate, ANN-based algorithms have been put forth. In this paper, a BLSTM-based SOC estimation technique is discussed, and results have been provided which prove the effectiveness of ANN-based algorithms in SOC estimation for HEV applications.

#### SOC Estimation Requirements in HEV Applications

The requirements for BMS change for different applications. The very nature of the functionality of HEVs makes SOC estimation very tricky compared to fully electric vehicles (EV). It is important to understand the difference in drive systems for EVs and HEVs.

EVs have relatively simple drive systems compared to HEVs. In an EV, the power from the battery pack is supplied to the motors through an inverter. Figure 2 shows a distributed multi motor electric vehicle drive system [38]. The system consists of two DC motors driving the front and rear wheels through the front and rear differential. Regenerative charging can be achieved while braking. The power flow is quite simple while in operation: during acceleration, from the start, the energy flows from the battery to the motors and remains the same until it is reversed when regenerative braking occurs.

HEVs often have more complex drive systems compared to EVs. Figure 3 shows a simplified version of the drive system found in a Toyota Prius. Note that the system is now only a front wheel drive (FWD) or rear wheel drive (RWD) instead of an all-wheel drive (AWD), as shown in Figure 2. The need for a differential is negated using a planetary gear system, which helps transfer the power to the drive wheels from multiple sources. Because of the hybrid architecture, the power draw from the battery changes during various operating conditions.
During acceleration at the start, most of the workload is carried by the electric system because of the incredible torque provided by the electric motor. The power draw from the battery is the maximum during this phase. In most HEVs, the ICE is entirely shut down while starting from a complete stop.During normal conditions, the power draw from the battery is reduced massively, and power coming from the ICE is split to drive the generator and the wheels. The generator is in turn used to power the electric motors.During sudden changes to the vehicle’s momentum, i.e., sudden acceleration or deceleration, power from the battery is either drawn to support the ICE output or the electric motors are used as generators and the battery pack is charged while regenerative braking is performed.During charge condition, the battery pack can be charged using the ICE output to drive the generator. The battery charge levels are monitored to maintain a minimum level of charge.

Because of these fluctuating power draws from the battery, a far more accurate and sophisticated SOC estimator is required in HEV applications.

Hybrid electric vehicles require a more complex battery and battery management systems because of the extremely dynamic rate profiles seen in HEVs as compared to BEVs and personal electronics [37]. HEVs and BEVs also require high current in the magnitudes of 20C, which leads to the cell chemistry to never be in equilibrium, and this leads to the use of very robust SOC estimators. A simple SOC estimator such as Coulomb counting can be employed in Pes because of the almost constant current draw that a battery experiences in a PE environment, but this kind of SOC estimator is not preferred in HEVs because of the dynamic current draw from the battery.

A good SOC estimator has an advantage of prolonging the lifetime of the battery pack, since we can aggressively exploit the precise SOC estimate to control overcharging or undercharging. It also improves battery pack designs and leads to the use of less batteries, which ultimately saves on weight, size and price of the battery pack. So, a precise SOC estimator is not only required because of the challenges posed by HEV architectures but also has added benefits, which are essential in making this technology viable for practical use. 

## 2. Materials and Methods

A Panasonic 18650PF cell is used to collect data for this study. The data is available online provided by Dr. Phillip Kollmeyer at The University of Wisconsin-Madison. The cell has a rated capacity of 2700 mAh at 20 °C, and other vital battery specifications are listed in Table 1. The acquired battery datasets involve four dynamic tests—namely, US06, urban dynamometer driving schedule (UDDS), highway fuel efficiency test (HWFET) and LA92—that are used to simulate the power profile of battery packs in EVs and HEVs. A series of nine drive cycles are performed in the order Cycle 1, 2, 3 and 4, US06, HWFET, UDDS, LA92, Neural Network. Cycles 1 through 4 consist of a random mix of data from US06, HWFET, UDDS and LA92. The power profile of the drive cycles was calculated for a Ford F150 truck with a 35 kWh battery pack. This battery pack is scaled for a single 18650PF cell. Training and testing of the proposed network were completed using these four dynamic tests. The acquired datasets are trained and tested on Python v3.1 using Pycharm IDE. The computer used to run the software is a Windows workbench with an Intel Xeon processor and a 12 GB NVIDIA TitanXp GPU.

The main measurements taken from the datasheets are the voltage (V), current (I) and temperature (°C). As clearly observed from Figure 4d, battery discharge capacity decreases as temperature decreases. So, it is important to include temperature data. Temperature data of −10 °C, 0 °C, 10 °C and 25 °C were used while training and testing of the proposed model. The temperature is not constant but fluctuates within a range with an initial setpoint of the above-mentioned temperatures. This poses more challenge while estimating SOC but provides an accurate representation of a real-world scenario. The 25 °C temperature data in US06 conditions fluctuate within a range of 25 to 32.7 °C with an average temperature of 29 °C, the 10 °C data range from 33 to 10 °C with an average temperature of 15.92 °C, 0 °C data range from 12 to 0 °C with an average temperature of 6.9 °C and −10 °C data range from 6.6 to −10 °C with an average temperature of −0.1 °C. There are two temperatures provided in the original datasets, the battery case temperature at the middle of the battery measured using a thermocouple and the chamber temperature the battery is placed in. The chamber temperature is controlled using a 8 cu.ft thermal chamber. For this study, the battery case temperature is used, which varies over time. Figure 4c illustrates the fluctuations in the temperature data for the US06 condition. Within the original dataset, tests such as drive cycles were considered important, and the data were recorded every 0.1 s; i.e., we have a time step of 0.1 s during the discharge working condition. Other processes such as charging and pauses were considered secondary and have a lower data rate of 60 s. One can reduce the data rate to a constant 1 s in both conditions to lower the computational power required during testing and training. We can also up-sample the data to achieve a 0.1 s data rate for the charge working condition. In this paper, a constant data rate of 0.1 s is considered for greater accuracy while compromising on calculation costs.

As seen in Figure 4a,b, current and voltage fluctuate a lot within a given dataset. These massive fluctuations create issues while training the algorithm. Voltage, current and temperature are all measured in different scales, and these do not contribute equally, while model fitting and learning and create an unwanted bias. To overcome this issue, the input measurements are normalized between 0 and 1. Normalizing the datasets will not only eliminate the issue discussed above but also speed up the learning process of the algorithm. In this paper, the normalization is completed using a min–max scaler function.
(2)$$x_{scaled}=\frac{x\;\left(n\right)-\min\left(x\right)}{\max\;\left(x\right)-\min\left(x\right)}$$
where *n* is the *n*th measurement in the datasheet. $x_{scaled}$ is the normalized value in the range (0, 1).

### 2.1. Performance Metrics

The performance of the proposed network is evaluated using these performance metrics. 

#### 2.1.1. Mean Absolute Error

MAE is an arithmetic average of the absolute errors while comparing two separate outcomes defining the same process. In this case, we compare the actual (*y*_k_) vs. the predicted values ($\hat{y}$_k_).
(3)$$\mathrm{MAE}=\frac{1}{N}\sum (y_{k}\;-\;{\hat{y}}_{k})$$
where *N* is the total number of timesteps available. MAE serves as a perfect metric for interpretation of the network, since it is proven to be a more natural measure of average error.

#### 2.1.2. Root Mean Square Error

RMSE is the average of the squared errors between actual and predicted values. It is ideal for showing small variances and large outliers in errors as compared to Mean Square Error; since the errors are squared, RMSE exaggerates any large errors. The expression used to calculate RMSE is
(4)$$\mathrm{RMSE}=\sqrt{\frac{\sum {\left(y_{k}-{\hat{y}}_{k}\right)}^{2}}{N}}$$
where ${\hat{y}}_{k}$ is the predicted value, $y_{k}$ is the actual value and *N* is the total number of timesteps. RMSE tends to increase depending on the sample size. Since the sample sizes of the databases used in this study are approximately equal, values of RMSE can be compared across different databases.

## 3. Proposed Network Architecture

### 3.1. Long Short-Term Memory

LSTM was proposed by Hochreiter et al. as a new gradient-based method in 1997 to overcome the drawbacks of recurrent backpropagation algorithms [39]; it achieves this by using memory cells instead of hidden nodes. LSTMs use gate units that learn to open and close access to the constant error which eliminates vanishing or explosion of the gradient, as seen in back-propagation networks or general RNNs. The basic structure of an LSTM unit can be seen in Figure 5. LSTM units calculate values for all the gates involved: namely, forget gate (f_t_), input gate (i_t_), output gate (o_t_) and cell memory (c_t_) for every forward pass at time t. These calculations can be summarized as follows
(5)$$f_{t}\;=\;\sigma \;(W_{f}x_{t}\;+\;U_{f}a_{t-1}\;+\;b_{f})\;i_{t}\;=\;\sigma \;(W_{i}x_{t}\;+\;U_{i}a_{t-1}\;+\;b_{i})\;o_{t}\;=\;\sigma \;(W_{o}x_{t}\;+\;U_{o}a_{t-1}\;+\;b_{o})\;c_{t}=f_{t}\;•\;c_{t-1}+(i_{t}\;•\;{\tilde{c}}_{t})\;{\tilde{c}}_{t}\;=\;\tanh\;(W_{c}x_{t}\;+\;U_{c}h_{t-1}\;+\;b_{c})\;a_{t}\;=\;o_{t}\;•\tanh\;(c_{t})$$
where W, U and b are the weight matrices and bias parameter, which is learned during training the network. The sigmoid function (σ) is bound between 0 and 1 and hence is perfect to be used for forget, input and output gate calculations, since this can be interpreted within the LSTM unit as a “forgetting factor”. While training the network, if the value of the input gate or forget gate is close to 0, it will be interpreted as a non-essential input or non-essential previous memory and will be eliminated. 

In an LSTM unit, the forget gate, output gate and input gate depend on present input (x_t_) and previous output (previous activation) (a_t−1_); cell memory is influenced by the previous memory (c_t−1_), forget gate and input gate. The overall output (a_t_) considers all the gates and cell memory.

#### Bi-Directional LSTM

It is an extension of unidirectional LSTM which consists of a forward pass and a backward pass. Figure 6a shows the general structure of a stacked bi-directional LSTM, the structure of Bi-LSTM facilitates the network to have both backwards and forward information [18]. Bi-LSTMs achieve this by using two hidden layers that process input data in the forward and backward directions. These hidden sequences are then fed to the same output layer. Bi-LSTMs use these forward and backward sequences and update the output sequence using the following equations
(6)$$\vec{a_{t}}=\sigma \;\left(W_{x\vec{a}}+U_{\vec{a}}{\vec{a}}_{t+1}+b_{\vec{a}}\right)\;\overset{\leftarrow }{a_{t}}=\sigma \left(W_{x\overset{\leftarrow }{a}}+U_{\overset{\leftarrow }{a}}{\overset{\leftarrow }{a}}_{t+1}+b_{\overset{\leftarrow }{a}}\right)\;y_{t}=W_{\vec{a}y}{\vec{a}}_{t}+W_{\overset{\leftarrow }{a}y}{\overset{\leftarrow }{a}}_{t}+b_{y}$$
where *W*, *U* and *b* are the weight matrices and bias parameter. The backwards and forward layers are iterated by feeding the network form *t* = *N* to 1 for the backwards layer and *t* = 1 to *N* for the forward layer. As stated before, since the network has information regarding the previous and future sequences, it is ideal for use in state-of-charge estimations. 

### 3.2. Proposed Stacked Encoder–Decoder Bi-LSTM 

The proposed network uses a stacked bi-directional LSTM block in combination with the encoder–decoder network. Encoder–decoder architecture has been used in various applications such as language processing [40,41], trajectory estimation [42] and SOC estimation [19]. The encoder–decoder network works by feeding the input sequence to an encoder block and estimating the probability distribution of the *t*th sample of the output sequence (*s_t_*) using a decoder block. The structure of an encoder–decoder architecture is shown in Figure 7. The input sequence $u_{1},\ldots ,\;u_{N}$ is passed through the encoder block, which generates a cell state vector ($c_{N}$) after N recursive steps. The cell state vector consists of the hidden states summarized by the encoder block, which can be given as $c_{N}=m\left(h_{1},\ldots ,h_{N}\right)$. Since the recurrent blocks within the encoder–decoder network can be changed to other types of recurrent blocks such as SRNN, GRU or LSTMs [19], a Bi-LSTM block is used in this study. Since Bi-LSTMs are bi-directional the hidden weights generated by the encoder block is a concatenation of forward ($fh_{i}$) and backward ($bh_{i}$) hidden states. The state vector can now be represented as $c_{N}=m\left((fh_{1},bh_{1}\right)\ldots ,\left(fh_{N},bh_{N}\right))$, where m is a non-linear function. The encoder block aims to model the conditional probability of the output sequence given the input sequence.
(7)$$P\left(s_{1},\ldots ,\;s_{N^{\prime }}\;|u_{1},\ldots ,\;u_{N}\right)=\overset{N^{\prime }}{\underset{n=1}{\prod }}P(s_{n}|c_{N},s_{1},\ldots ,\;s_{n-1})$$
where $s_{i}$ is the output sequence, $u_{i}$ is the input sequence and *N* is the number of time steps in the input data. The cell state vector of the encoder block is made to be the initial state of the decoder block, $c_{0}^{\prime }=c_{N}$, and the decoder generates a probability distribution for the nth sample of the output sequence given the decoder state of the previous sample ($c_{n^{\prime }-1}$) and the previous sample of the output sequence ($s_{n-1}$).
(8)$$P\left(s_{1},\ldots ,\;s_{N^{\prime }}\;|u_{1},\ldots ,\;u_{N}\right)=\overset{N^{\prime }}{\underset{n=1}{\prod }}P(s_{n}|{c^{\prime }}_{n^{\prime }-1},s_{n-1})$$
where $N^{\prime }$ represents the number of time steps in the output sequence. Both the encoder and decoder blocks are trained together, and the decoder outputs the target sequence given the input sequence. The flow of the proposed algorithm is shown in Figure 6b. The hidden states of the stacked layer are fed into the encoder input. Then, the encoder processes the input sequence and forms the cell state network to be used by the decoder to estimate the output sequence. The decoder uses the estimated previous sample (${s^{\prime }}_{n-1}$) to estimate the present sample because of the unavailability of true sample values to the decoder. This is the main limitation of using the proposed architecture. Since the decoder block does not have access to the true sample values and uses an estimated previous sample value to estimate the current sample, there is a chance of error propagation to occur. The error propagation can be limited by using a beam search algorithm [42].

The input and target output sequences are divided into $\frac{N}{L}$ sequences with each sequence of length L and fed into the network. The sequence length (*L*) is set to optimize the estimation accuracy of the network. To overcome the issue of vanishing gradients, a rectified linear unit (*ReLU*) activation function is used, and the output of the dense layer at the end can be given as
(9)$$SoC_{i}=ReLU\left(w^{N}\cdot h_{D}^{i}+b\right)$$
where *w* is the weight matrix and b is the bias associated with the fully connected dense layer. Training the model consists of a forward pass and a backwards pass. The network creates an estimated sequence and calculates the loss function. Mean Square Error was used as the loss function in this study, which is given as the average of the squared difference between the actual and estimated values. Total loss is sent backwards through back propagation to update the weights and biases accordingly. Back propagation is performed using an adaptive moment optimizer also known as Adam optimizer. During testing, no back propagation is performed, as the network has finished learning and the weights and biases are not updated. Furthermore, to avoid overfitting, a dropout layer can be implemented in between the Bi-LSTM layers [19,43].

The proposed network utilizes the features from stacked Bi-LSTM and ED architecture.

## 4. Experimental Results and Discussion

Stacked Bi-LSTM and encoder–decoder Bi-LSTM (ED Bi-LSTM) are used to compare the functionality with the stacked encoder–decoder network (SED). The networks are tested across different temperature ranges (25 °C, 10 °C, 0 °C, −10 °C) of the datasets available. The hyperparameters of all the networks were made constant, and no hyperparameter optimization was performed prior to testing. Hyperparameter optimization can be completed as part of future development of the proposed network. Computational time was given priority while testing the networks; all the hyperparameters are selected in a way that reduces the time taken to train and test the network. The SED Bi-LSTM network takes the longest to train compared to the other two networks, which is due to the deep architecture of the network. On average, the proposed network takes 1.5 to 2 h longer compared to stacked Bi-LSTM and 1 to 1.5 h longer compared to ED Bi-LSTM. Measures were taken to accelerate the training process in the SED network by normalizing the input data and training all the internal blocks parallelly. The number of iterations has been limited to under 1000, and the number of trainable parameters that affect the network width has been limited to less than 50,000. To further emphasize the advantages of the proposed network, a smaller number of trainable parameters are used in the SED network compared to ED Bi-LSTM and stacked Bi-LSTM. Figure 8 and Figure 9 show the SOC prediction comparison of different networks. All the evaluation metrics are computed by an error calculation function after testing the network. To check the proper functionality of the error calculation function, the software is run through a known set of estimated values whose error metrics are previously calculated.

Table 2 shows the effects of varying the depth of the algorithm. The depth of the algorithm can be changed by changing the number of Bi-LSTM layers within the stacked and encoder–decoder blocks. It is very evident for the results that the algorithm performs well when two Bi-LSTM layers are used throughout the architecture. Further increasing the model depth to three Bi-LSTM layers affects the algorithm negatively and the accuracy drops; this can be attributed to the vanishing gradients. The depth of the algorithm was set to two layers for further evaluations within this study. Table 3 shows the performance metrics for all the networks compared in the study at different temperatures. It is evident that the proposed network performs better in all the conditions compared to stacked Bi-LSTM and ED Bi-LSTM. The MAE and RMSE are lower in all the cases for the proposed network. The lowest MSE of less than 1% was observed under UDDS condition while using the SED network. The error metrics increase in US06 conditions due to the high discharge currents involved, but the overall error for the SED network is still lower than that in the other two networks. The highest MAE of 1.97% and RMSE of 2.7% is observed in US06 condition at 0 °C. 

From Figure 8 and Figure 9, stacked Bi-LSTM performs the worst overall and takes the least amount of time to train among the networks tested. The SOC prediction error shown in the SOC prediction error plots for all the networks is the least for the UDDS condition at 25 °C with an overall error within ±0.05 of the actual value at every timestep and the most for US06 conditions at 0 °C with an overall error within ±0.15 of the actual value at every timestep. The SED network performs the best, since the network depth is increased with the stacked layers and the encoder–decoder setup provides more accurate predictions in many-to-one sequence scenarios. The SOC prediction error plots show that the estimation error was the highest for stacked Bi-LSTM in almost all the cases. The effectiveness of the encoder–decoder architecture is enhanced using stacked layers in the SED network and is reflected in the HWFET condition at 25 °C; stacked Bi-LSTM performs slightly better than ED Bi-LSTM in this condition, but the SED network outperforms both, even though the standalone ED architecture fails to do so. The lowest MAE of 0.62% and RMSE of 0.86% is observed in HWFET conditions at 25 °C. Overall, under UDDS conditions, the SED network performs the best with MAE and RMSE of less than 1.5%. Under HWFET and LA92 conditions, the MAE and RMSE are less than 1.8% and 2%, respectively. Under the US06 condition, the MAE and RMSE is less than 2.8%, but it is relatively higher compared to other conditions because of the unstable cell equilibrium due to the high discharge current. This affects the model prediction, since current, voltage and temperature characteristics are used to predict the SOC at a given timestep. LSTM layers and the ReLu activation function are used to overcome the issue of vanishing gradients, and limiting the number of parameters helps in reducing the likelihood of overfitting. From the results discussed above, it can be concluded that the SED network performs the best, and the use of stacked layers combined with encoder–decoder architecture leads to an improvement in performance throughout all the conditions tested in this study.

## 5. Conclusions

In this paper, a new sequence to sequence deep learning algorithm is proposed to improve on pre-existing SOC estimation techniques. A stacked encoder–decoder algorithm is introduced for hybrid electric vehicle applications. The major contributions are as follows. Firstly, the proposed algorithm improves on previously established encoder–decoder architecture when SOC is estimated at varying temperatures. Secondly, the SED algorithm can learn from the measured data of lithium-ion batteries and directly estimate the SOC at varying temperatures. The algorithm analyzes the data sequentially and generates an SOC sequence based of the context of the input sequences. Thirdly, data processing techniques are implemented to reduce the time taken to train the algorithm and lower the computational load. The network depth study shows that the network performs the best when two Bi-LSTM layers are used and worse when three Bi-LSTM layers are used in all the blocks within the network. This also shows the limitation of deep neural networks, which are prone to vanishing gradients when the network becomes too deep. The use of encoder–decoder architecture helps in reducing the training time because the encoder and decoder blocks are trained simultaneously. Experimental results have shown that the proposed algorithm performs better than standalone encoder–decoder architecture and stacked bi-directional LSTM architecture. An MAE as low as 0.62% was observed while estimating the SOC at varying temperature, which proves the practicality of the proposed algorithm. Since the proposed network is not a model-based network, it can be implemented in various other applications for real-time SOC estimation, provided that training data are made available for the network. The SOC can be estimated while charging with a supercharger or household outlet. The proposed network can be used in an EV with a solar panel setup, since the battery pack undergoes similar conditions as the HEV environment where the battery sustains short charging and discharging conditions. The algorithm can be further improved by implementing EKF based hyper-parameter estimation techniques for hyper-parameter optimization and using techniques such as the beam search algorithm to overcome the inherit drawback of using an encoder–decoder structure. Finally, because of the very accurate SOC estimation obtained, one can aggressively exploit it to control over-charging or under-charging the battery pack. This helps in reducing the number of batteries required within the battery pack and ultimately contributes to reducing the cost of the battery pack. In conclusion, the proposed algorithm improves on the existing SOC estimation techniques and is a good choice for EV and HEV BMS applications. 

## Figures and Tables

**Figure 1 micromachines-13-01397-f001:**
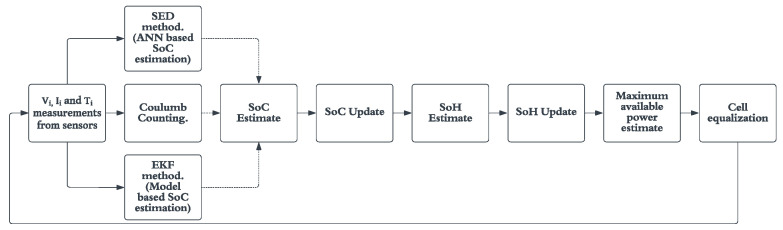
General BMS task flow.

**Figure 2 micromachines-13-01397-f002:**
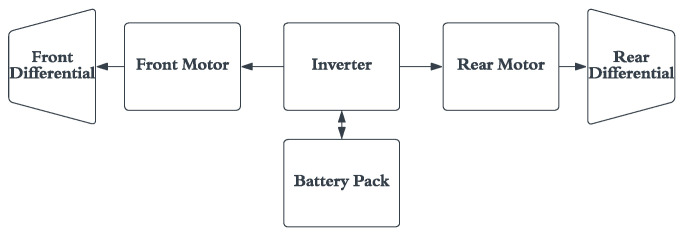
Drive system of a fully electric vehicle.

**Figure 3 micromachines-13-01397-f003:**
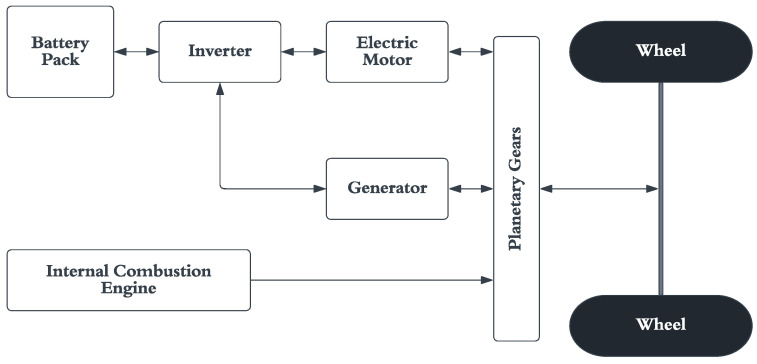
Drive system of a hybrid electric vehicle.

**Figure 4 micromachines-13-01397-f004:**
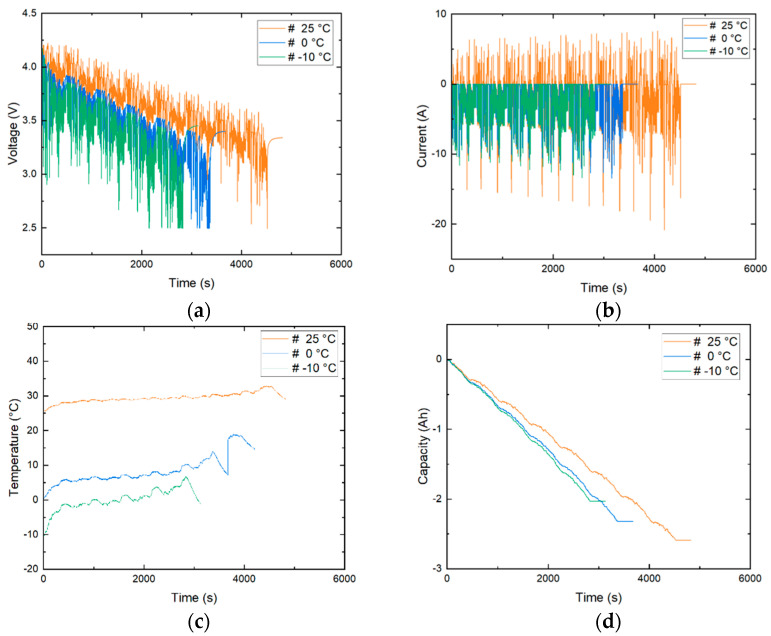
US06 test data at 25 °C, 0 °C and −10 °C. (**a**) Voltage, (**b**) Current, (**c**) Battery temperature, (**d**) Capacity.

**Figure 5 micromachines-13-01397-f005:**
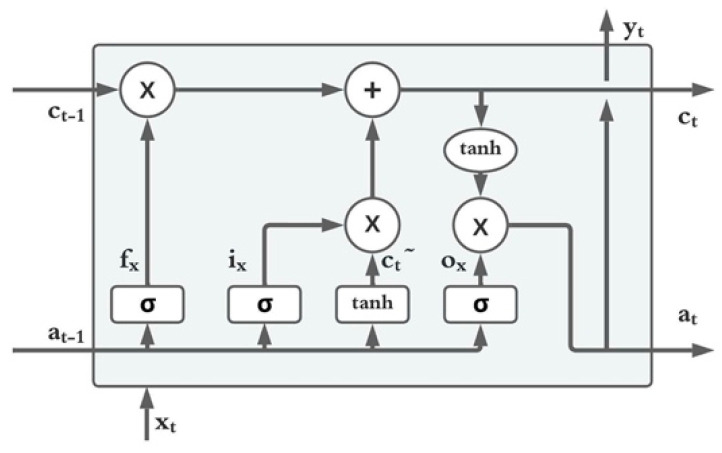
Structure of a LSTM unit.

**Figure 6 micromachines-13-01397-f006:**
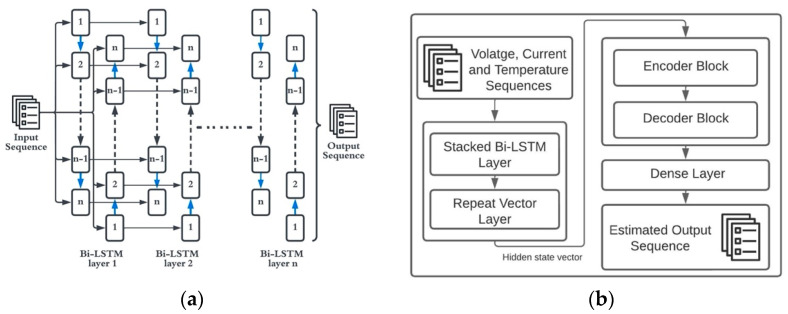
Schematic Diagram (**a**) Stacked Bi–LSTM structure. (**b**) Stacked Encoder–Decoder Architecture.

**Figure 7 micromachines-13-01397-f007:**
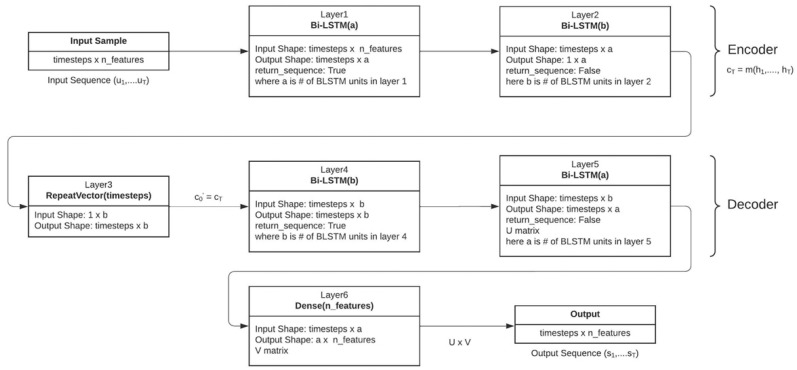
Encoder–Decoder Architecture.

**Figure 8 micromachines-13-01397-f008:**
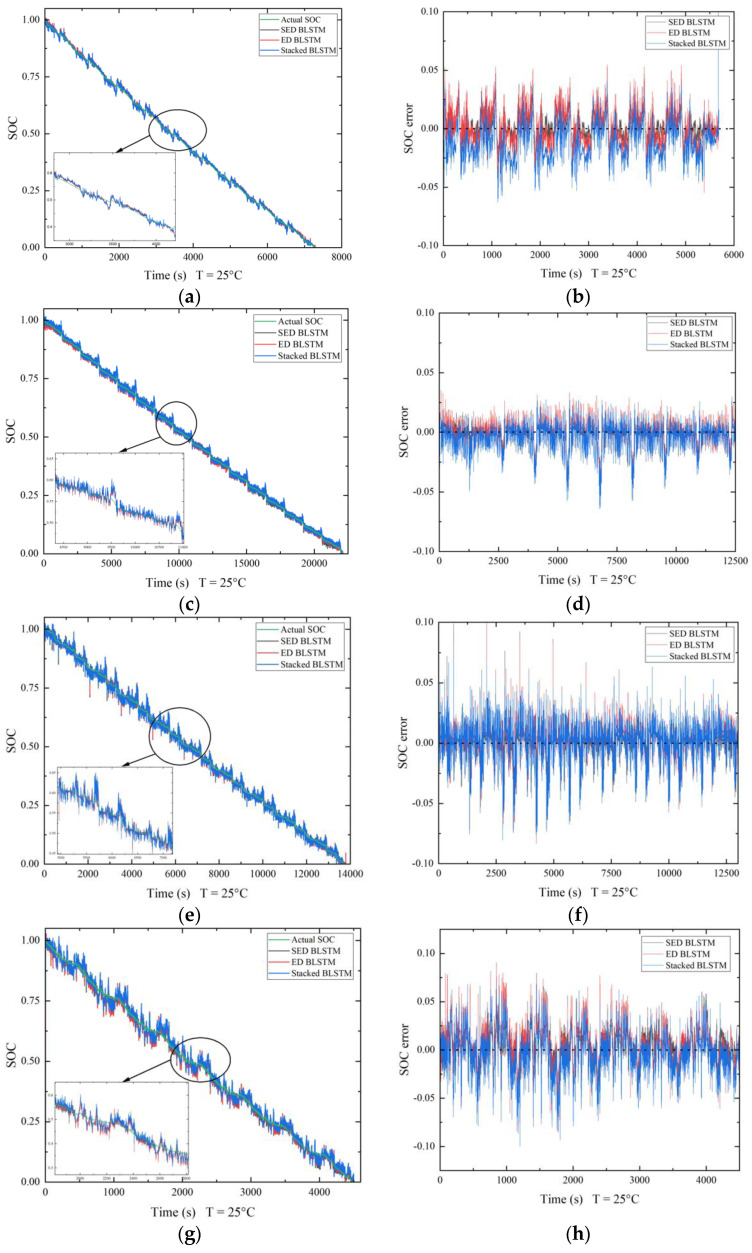
Predicted SOC comparison and SOC prediction error at 25 °C. (**a**,**b**) HWFET. (**c**,**d**) UDDS. (**e**,**f**) LA92. (**g**,**h**) US06.

**Figure 9 micromachines-13-01397-f009:**
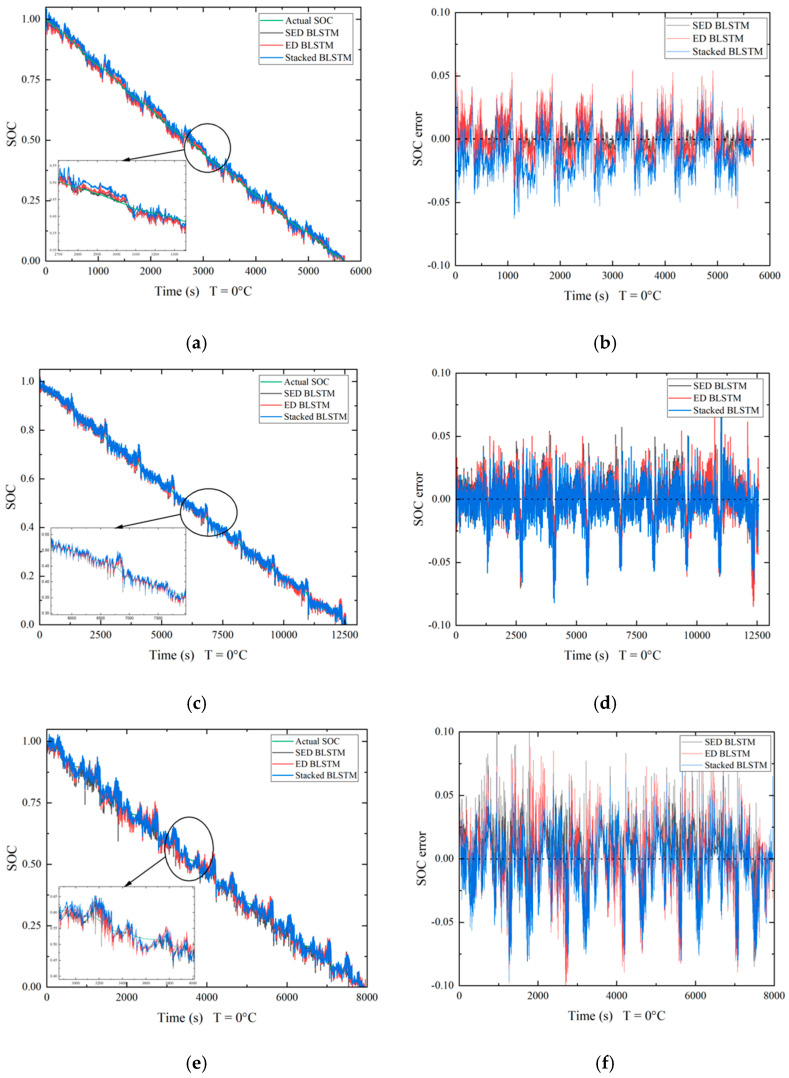
Predicted SOC comparison and SOC prediction error at 0 °C. (**a**,**b**) HWFET. (**c**,**d**) UDDS. (**e**,**f**) LA92. (**g**,**h**) US06.

**Table 1 micromachines-13-01397-t001:** 18650PF battery specifications.

Rated Capacity		2700 mAh	2615 mAh
Capacity	Minimum	2750 mAh	2665 mAh
	Typical	2900 mAh	2810 mAh
Nominal Voltage		3.6 V	
Charging	Voltage	4.20 V	4.15 V
	Current	0.5 C	
Energy Density	Volumetric	577 Wh/L	559 Wh/L
	Gravimetric	207 Wh/kg	200 Wh/kg

**Table 2 micromachines-13-01397-t002:** Comparison of different number of bi-directional LSTM layers within the stacked and encoder–decoder blocks.

Bi-LSTM Layers	Metrics (%)	Temp (°C)			
		25	10	0	−10
1 layer	MAE	0.7226	1.0292	1.2268	1.4880
	RMSE	1.0019	1.2988	1.5101	1.9876
2 layers	MAE	0.6229	0.9957	1.1066	1.2021
	RMSE	0.8615	1.2832	1.3884	1.7240
3 layers	MAE	0.8268	2.3668	1.3155	1.5877
	RMSE	1.0365	2.9332	1.6815	2.0852

**Table 3 micromachines-13-01397-t003:** Performance metrics at different temperatures.

Network Model	Temperature (°C)	UDDS MAE (%)	RMSE (%)	HWFET MAE (%)	RMSE (%)	US06 MAE (%)	RMSE (%)	LA92 MAE (%)	RMSE (%)
SED	−10	0.7768	1.2233	1.2021	1.7240	1.2289	1.8075	0.6843	1.3100
	0	1.0502	1.4381	1.1066	1.3884	1.9743	2.7022	1.6693	2.0993
	10	0.8829	1.2134	0.9957	1.2832	1.9457	2.5715	1.1107	1.6442
	25	0.6478	0.9278	0.6229	0.8615	1.3780	1.8510	0.9508	1.3381
ED	−10	1.4543	2.0943	1.5284	1.9751	2.5922	3.5435	2.5011	3.3832
	0	1.0195	1.4330	1.1760	1.4570	2.4400	3.2085	1.9554	2.5138
	10	0.9843	1.3669	1.0656	1.3695	2.5067	3.2818	1.6248	2.1145
	25	0.6819	0.9543	0.7375	0.9531	1.6231	2.1357	1.0169	1.3792
Stacked	−10	1.5341	2.2080	1.7923	2.2359	2.9908	4.0846	2.7319	3.5576
	0	1.0294	1.4654	1.5315	1.8561	2.5751	3.3895	1.7562	2.3026
	10	0.9640	1.3942	1.0493	1.3720	2.5767	3.3570	1.4743	2.0378
	25	0.6827	1.0098	1.7187	1.0461	1.6089	2.1293	1.0363	1.4184

## Data Availability

All the data used to test the functionality of the algorithm in this study are open-source data provided by Mendeley data. Link to data: https://data.mendeley.com/datasets/wykht8y7tg/1 (accessed on 10 November 2021).

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
