# Peer review of "State of Charge Estimation of Lithium-Ion Batteries Using Stacked Encoder–Decoder Bi-Directional LSTM for EV and HEV Applications"

_micromachines, 2022, doi:10.3390/mi13091397_

Round 1
Reviewer 1 Report
Dear Editors and Authors:
The reviewer is hereby sending you the review report.
Please find the attached file to this email.
The reviewer highly appreciates your work and contributions.
Thank you
Sincerely yours,
The reviewer

Reviewer 2 Report
This manuscript presents an interesting battery State of Charge estimation method. The proposed design could show lower errors than the two existing methods so-called stacked and ED BLSTM. The comparison was done under four different initial temperatures using none selected drive cycles. Overall the paper is well written, but there are minor errors to be amended before it can be considered for publication. Please find my specific comments below:
1. Kindly refrain from mentioning certain car brands, especially in the Abstract section. I feel that the first sentence could be removed without losing the whole message of the Abstract.
2. Table 1 is not mentioned in the main text.
3. Line 259: there is no figure 2(d) in the manuscript. It should be Figure 4(d).
4. Lines 261-267 and Fig. 4:
a. What kind of initial temperature conditioning methods were used in this study?
b. As shown in Fig. 4(c), the -10°C, 0°C, 10°C, and 25°C temperatures were only the initial setpoint, correct?
c. Is there any importance in presenting the average temperature values?
d. I could not comprehend why there was a sudden temperature drop, especially towards the end of the cycle. Except if there is some sort of temperature control. Could the authors please explain this issue?
5. Lines 435-436 are pretty interesting, especially since the values are larger than a single-layer case (Table 2). Perhaps it could be added to the Abstract and/or Conclusions.
6. Line 445: there should not be any figures with no captions in the manuscript. They should be combined with the remaining Figures 8 (line 472).
7. Figs. 8-9 insets are too small.
